# Kilonova Emission and Heavy Element Nucleosynthesis

**Elena Pian** [1,2]

1   Italian National Institute of Astrophysics (INAF), Astrophysics and Space Science Observatory, Via P. Gobetti 101, 40129 Bologna, Italy; elena.pian@inaf.it
2   Astrophysics Research Institute, Liverpool John Moores University, IC2, Liverpool Science Park, 146 Brownlow Hill, Liverpool L3 5RF, UK

**Abstract:** The binary neutron star merger observed and localized on 17 August 2017 by the LIGO and Virgo gravitational interferometers and by numerous telescopes on the ground and in orbit linked in an unambiguous way the coalescence of double neutron stars with the formation of a relativistic outflow (short gamma-ray burst GRB170817A) and of a thermal radioactive source (kilonova). The vicinity of the event (40 Mpc) made it possible to monitor the electromagnetic counterpart in detail at all wavelengths and to map its close environment in the outskirts of the lenticular galaxy NGC 4993. Radio VLBI images of GRB170817A allowed the first direct detection of superluminal motion in a GRB afterglow, pointing to a collimated ultra-relativistic jet rather than to a quasi-isotropically, mildly relativistically expanding source. The accurate spectroscopy of the kilonova at ultraviolet-to-infrared wavelengths with the X-Shooter spectrograph of the ESO Very Large Telescope showed the long-sought-after signature of rapid neutron capture process (in short: r-process) nucleosynthesis. Kilonova detection makes gravitational wave sources optimal tracers of heavy element formation sites.

**Keywords:** nucleosynthesis; neutron stars; gamma-ray bursts; gravitational waves

## 1. Mergers of Binary Compact Star Systems

Binary systems of neutron stars and black holes, including hybrid systems composed of a neutron star and a black hole, form in the universe in various ways. One of these predicts that both members of a binary massive star system undergo core-collapse supernova explosion, each leaving behind a degenerate remnant: a neutron star or black hole. Stellar cores larger than about 100 $M_\odot$ may implode directly to a black hole without undergoing supernova. If the binary compact star system thus formed is not disrupted by the second collapse, it survives some hundreds of millions of years, during which the orbital motion of the members causes the system to loose energy via gravitational radiation and eventually to coalesce. If one member is a black hole, the remnant of the merger is naturally always a black hole, while if both members are neutron stars, the final stable black hole configuration may be preceded by a brief (tens to hundreds of milliseconds) metastable supra-massive neutron star remnant phase. The regime of strong gravity (large masses confined in a small volume and in tight orbits) and the precise orbital behavior that characterize these sources make them among the loudest and best identifiable sources of gravitational radiation for state-of-the-art laser interferometers in the 100–1000 Hz range, where the detectors have maximum sensitivity [1].

The formation rates of these systems are poorly constrained, as they depend on star formation and evolution in a way that makes them rather uncertain [2–4]. The most illustrious example of these binary collapsed star systems in our galaxy is PSR1913+16, a pulsar that orbits around an unseen companion with a period of about 8 h, that decreases secularly according to gravitational radiation loss prediction [5,6]. The 1974 report of this system and of its shrinking orbit, which earned the Nobel Prize in 1993 for its authors Russell Hulse and Joseph Taylor, was followed by many other similar detections. To date,

about 20 binary neutron star systems have been detected in the galaxy [7–9], but no binary black hole system nor neutron star plus black hole system has been detected.

## 2. Aftermath of a Binary Merger Containing a Neutron Star

While it is unlikely, owing to relatively low rates, that a binary compact star merger takes place in the galaxy in our lifetime, these events occur frequently in the universe. Binary stellar-size black hole systems are thought to be electromagnetically silent, which makes the identification of their host galaxies difficult, since the error areas provided by current gravitational interferometers project up to several hundreds of square degrees in the sky. These sky areas contain high numbers of galaxies, depending on limiting brightness. As an example, at a distance of 200 Mpc, in an error box of $\sim$20 deg$^2$, there may be approximately two dozen galaxies that have a luminosity comparable to the Milky Way [10].

On the other hand, the mergers of binary systems, where at least one member is a neutron star, are accompanied by multi-wavelength transients. Therefore, these systems can be detected through their multi-messenger variable signals from external galaxies, as predicted long before the advent of the gravitational laser interferometers Advanced LIGO and Virgo [11–13]. The expected electromagnetic signals include: a prompt short-duration gamma-ray burst (GRB)—order of 2 s or less, see [14] for the classical distinction between long and short GRBs, barring extended emission in short GRBs [15]—produced by a mildly relativistic cocoon and/or highly relativistic outflow collimated by the strong magnetic field of the merger compact remnant or anchored on its promptly developed accretion disk; a GRB multi-wavelength afterglow of non-thermal origin (synchrotron radiation), which rises hours to days after the merger, depending on the viewing angle; a thermal source of radioactive nature due to the formation of heavy elements, the so-called kilonova, which forms milliseconds after the merger, rises in days and fades rapidly (faster than a supernova); low-level non-thermal radio and X-ray emissions resulting from the interaction of the kilonova ejecta with the circum-binary medium, peaking years after the merger, a.k.a. kilonova afterglow.

## 3. Kilonovae

As their name suggests, kilonovae are transient sources about three orders of magnitude more optically luminous than novae; that is, they are expected to reach a maximum luminosity of $\sim$10$^{40}$–10$^{41}$ erg s$^{-1}$. They have been predicted to exist as the aftermath of the merger of binary systems of two neutron stars or of a neutron star and a black hole [16]. Upon merger, a fraction ($\sim$10$^{-3}$–10$^{-2}$ M$_\odot$) of neutron-rich material comes unbound and decompresses rapidly. The results of hydrodynamic and full-network calculations [17] and of 3D simulations [12] have shown that rapid neutron capture (a.k.a. "r-process") nucleosynthesis takes place in this environment and produces sufficient material to explain most of the r-process nuclei in the galaxy. Binary neutron star mergers may in fact be responsible for a large fraction, if not all, of the synthesis of r-process elements in the universe, as the neutron and energy densities attained in supernovae may not be sufficient to produce these elements [18–22]. An exception may be the energetic supernovae connected with GRBs, where, in a collapsar scenario, the accretion disk could host r-process nucleosynthesis [23–25].

Kilonovae, like novae and supernovae, are essentially thermal sources powered by the radioactive decay of hundreds of unstable isotopes of neutron-rich elements with an atomic weight greater than that of iron, formed via r-process after the two compact stars coalescence. Unlike supernovae, whose light is almost entirely produced by the radioactive element $^{56}$Ni, kilonova light curves are due to the convolution of the radioactive decay of comparably abundant isotopes. This makes it difficult to disentangle the individual radioactive species present in the ejecta solely based on the analysis of the light curves, although their slopes may provide important constraints (see Section 6). More accurate diagnostics on chemical composition should come from spectroscopy.

Two components are thought to produce the kilonova spectral continuum: one is due to the dynamical ejecta, consisting of the mostly neutron-rich material (i.e., with relatively low ratio between electrons and nucleons, equal to the ratio between protons and nucleons, symbolized by the $Y_e$ parameter) resulting from the tidal disruption of both neutron stars and distributed in a thick accretion disk on the plane of the merger. The high opacity of the elements in this material, thought to be dominated by lanthanides, suppresses the continuum at the bluer wavelengths, thus making this spectral component appear relatively "red" [26,27]. The other component forms occasionally in a post-merger phase, along the polar direction, when a long-lived (tens of milliseconds) neutron star remnant produces a neutrino wind that lowers the neutron fraction and favours the formation of more neutron-poor elements (i.e., having high $Y_e$). With their lower opacity, these elements are less effective in suppressing the bluer wavelengths.

Kilonova continuum emissions have a spectral maximum in the wavelength range 3000–20,000 Å, where the neutral and ionized atoms of the freshly formed unstable isotopes imprint absorption lines. Spectroscopy in this range thus offers a diagnostic of the nature and abundance of these elements. No firm identification of a kilonova had occurred prior to the gravitational event of 17 August 2017 (see Section 6). However, infrared excesses had been detected superimposed on the light curve of the afterglow of short GRB130603B, which would be consistent with kilonova emission models [28,29]. Similarly suggestive reports were made for other nearby GRBs [30,31].

## 4. The Gravitational Event of 17 August 2017

On 17 August 2017, the two LIGO detectors revealed a signal, GW170817, that was consistent with being generated by a binary neutron star inspiral and merger [32]. The Virgo non-detection helped reduce the positional uncertainty of the source in the sky. Simultaneously, the Gamma-ray Burst Monitor onboard *Fermi* and the *INTEGRAL* ACS detected a 2-second-duration GRB from a sky area co-located with the error area associated with the gravitational signal [33–35]. Interestingly, the start time of this GRB lags the merger gravitational signal by 1.7 s, a time delay that may be related to the formation of the GRB engine or the jet acceleration process, or to a simple geometric effect, i.e., due to the gamma radiation emitted from the jet axis, thus delayed with respect to the gravitational wave [36].

A previously unknown point-like optical source—named AT2017gfo—was detected 11 h after the gravitational signal at a projected separation of about 10 arcseconds (i.e., ∼2 kpc) from the center of galaxy NGC 4993, located in the LIGO error area, at the known distance of 40 Mpc. This is compatible with the distance estimated with the gravitational data based on the "standard siren" assumption [32,37,38]. The near-infrared/optical spectrum of this source identified it as a kilonova, as detailed in Section 3. It showed broad absorption lines, the tell-tale sign of atoms heavier than iron present in the ejecta, formed via rapid neutron capture nucleosynthesis in the expanding, neutron-rich plasma released after the tidal disruption of the two neutron stars [39–41]. The next two sections contain an account of the major results obtained with observations of the counterpart of GW170817 across the electromagnetic spectrum, with an emphasis on the kilonova. A detailed review of the electromagnetic event observations can be found in [42–44].

## 5. The Non-Thermal Source Associated with GW170817

The short GRB detected simultaneously and co-spatially with GW170817 by the *INTEGRAL*/SPI–ACS [35] and *Fermi*/GBM [34], GRB170817A, with a total isotropic energy of ∼$3 \times 10^{46}$ erg, was under-energetic with respect to the rest of the short-GRB population, whose isotropic energies average around $10^{51}$ erg [45,46]. The relatively large viewing angle, suggested by the orientation of the gravitational wave source [32], may be responsible for the lack of relativistic beaming of the gamma-rays, and for the intrinsic weakness of the GRB signal, which would have indeed gone undetected were the source located at the typical distance of the majority of GRBs (z∼1). No GRB afterglow at X-ray or radio wavelengths was detected until about 10 days after explosion, which is again consistent

with expectations for a significantly off-axis view to the GRB jet [47,48]. Radio VLBI interferometry led Mooley et al. [49] to detect superluminal motion of the source, pointing to a jet blob moving relativistically. Ghirlanda et al. [50] then determined that the blob remains unresolved at radio wavelengths, indicating that the jet has a structured profile, i.e., with the Lorentz factor varying as a function of viewing angle, rather than a wide-angle, mildly relativistic quasi-isotropic outflow (choked jet or cocoon). The estimate of the jet viewing angle from the VLBI monitoring, ∼20 degrees, is consistent with that of the symmetry axis of the gravitational source [32].

## 6. The Kilonova AT2017gfo Associated with GW170817

Analysis of the ultraviolet-to-infrared light curve of AT2017gfo shows that its slope is consistent with the radioactive decay of neutron star ejecta with electron fraction $Y_e \lesssim 0.3$, which provides strong evidence for an r-process origin of the electromagnetic emissions in this band [51].

Owing to solar constraints, spectroscopic observations of the kilonova AT2017gfo started at the largest telescopes around the world many hours after detection of the optical counterpart, except at the Magellan telescope, where spectra could be acquired less than an hour after the optical detection; these showed a black body peaking shortward of the detector wavelength range, i.e., with a temperature larger than 5000 K [52], indicating that the source may have had its maximum emission in the ultraviolet range during its very early phase.

Ten very high signal-to-noise ratio spectra of AT2017gfo, with excellent wavelength coverage (3000–22,200 Å), were obtained with the ESO Very Large Telescope equipped with the X-Shooter spectrograph [39,40]. The earliest of these, acquired 1.5 days after the merger, describes a black body only weakly modified by absorption; the later spectra, as the ejecta become transparent, deviate increasingly from a thermal law and show more absorption lines, whose profiles are broadened by large photospheric velocities of the order of 0.1–0.2$c$. This causes blending of the lines, which, together with their blueshift and potentially very high number (large atomic number elements can have millions of transitions), makes their identification arduous. Additional difficulty arises from our poor knowledge of the line opacities, i.e., the probability of the transitions and their optical depth [53–55]. Examples of the interplay of the more neutron-rich and more neutron-poor model components (see Section 3) in reproducing the observed continuum at early epochs after merger can be found in [39,56–58].

A preliminary attempt at Cs I and Te I absorption identification [40] was disputed by Watson et al. [59], who proposed instead that the strong, broadened P-Cygni absorption feature observed at 1.5 days around 8000 Å corresponds to a transition of once-ionized strontium (Sr II), blue-shifted along the line of sight owing to the approaching photosphere. Since the Sr element is close to the first peak of the r-process elemental abundance distribution, which includes the neutron-poorest ones [60,61], its identification may imply that low-neutron-content atoms prevail in the ejecta, i.e., the kilonova emission we observe may be dominated by post-merger ejecta, favouring a long-lived neutron star as a remnant (see also [62,63]). Upper limits on platinum and gold amounts indicate masses lower than a few $10^{-3}$ and $10^{-2}$ M$_\odot$, respectively [54]. The *Spitzer* mid-infrared spectra obtained in nebular phase at 43 and 74 days after merger [64,65] suggest, depending on assumptions, the presence of Se, W, Os, Rh and Ce atoms in low ionization states [66].

In the wake of the first kilonova detection, its ultraviolet-to-infrared light curves were systematically compared to those of the counterparts of low-redshift short GRBs to explore possible analogies and differences of this thermal component in the various cases [67–69]. Considerable diversity is observed in the kilonova properties that cannot be ascribed only to the viewing angle. It appears that, when evidence of a kilonova is found in an afterglow, its red component may be similar in luminosity to that of AT2017gfo, while the blue component may be substantially brighter.

The predicted long-term aftermath of the kilonova, caused by the interaction of the merger ejecta with the external medium (kilonova afterglow), may have been detected in X-rays with *Chandra* about 3.5 years after binary merger [70].

## 7. The Host Galaxy of GW170817

The binary neutron star system that produced GW170817 was hosted by a lenticular galaxy, NGC 4993, at $z = 0.009783$ (40 Mpc), that was accurately studied with, among others, the *Hubble Space Telescope*, *Chandra* and the MUSE instrument on the ESO Very Large Telescope [38]. Imaging of the continuum with the *Hubble Space Telescope* suggests evidence for large, face-on spiral shells, while MUSE integral-field spectroscopy revealed edge-on spiral features emerging from the distribution of nebular line emission. The *Chandra* imaging points to the presence of mild nuclear activity. All this suggests that NGC 4993 has undergone a relatively recent merger ($\sim$1 Gyr), which may have fueled the weak active nucleus. The total stellar mass of the galaxy was estimated to be $M_* \sim 1.4 \times 10^{11}$ $M_\odot$ based on a fit to the integrated spectral energy distribution of the galaxy. This, together with little or no ongoing star formation, is consistent with the properties of host galaxies of short GRBs [46].

At the location of AT2017gfo, no globular or young stellar clusters more massive than a few thousand solar masses is detected with the *Hubble Space Telescope*, which is many standard deviations below the peak of the globular-cluster mass function of the host galaxy, strongly suggesting that GW170817 did not form and merge in a globular cluster [71]. The population in the vicinity of the merger is predominantly old, with $\lesssim$1% of the light due to a population younger than about 500 Myr [38].

Statistical tools based on binary stellar evolution simulations and binary population and spectral synthesis codes can prove effective in guiding a general estimate of the lifetime of the merging binary system and of the basic properties of its members in the present and future detected cases, provided due account is taken of the many involved assumptions and uncertainties [72].

## 8. Conclusions and Future Prospects

The gravitational waves detected from GW170817 and the observation of its counterpart over the electromagnetic spectrum represent a textbook astrophysical case of long-standing multiple hypotheses confirmation. Direct detection of gravitational radiation from a binary neutron star inspiral was anticipated after the indirect evidence of orbital shrinking provided by long-term monitoring at radio wavelengths of PSR1913+16; short GRBs were long suspected to be parented by binary neutron star mergers, based on their sub-second duration. A thermal source powered by radioactive decay of unstable isotopes of heavy elements, generated via r-process in the neutron-rich medium surrounding the merger, was considered to be a virtually unavoidable aftermath of neutron-star binary coalescence.

The closeness of the GW170817 event limited the number of galaxies that could potentially host its progenitor, facilitating the search with optical telescopes, and, together with the orientation of its symmetry axis (about 30 degrees away from our line of sight), largely favoured the detection of all three phenomena: GW signal, GRB, and kilonova. The limits on the ejecta mass of the kilonova, few hundreths of a solar mass [39,73], together with estimated rates of binary neutron star mergers, directly confirms earlier purely theoretical or indirect observational conclusions that double neutron star mergers are indeed a major site of cosmic nucleosynthesis [51,74].

Several candidates for binary neutron star mergers or neutron star and black hole mergers were detected during the third observing run of the LIGO/Virgo/Kagra interferometers in 2019–2020 [1,75–77]. However, perhaps owing to the relatively large estimated distance and size of the error areas in the sky, no electromagnetic counterpart could be solidly detected, with the possible exception of a gamma-ray signal reported by the *INTEGRAL* SPI-ACS in connection with GW190425 [78]. The results of the third observing run

constrained the cosmic rates of binary neutron star mergers and neutron star/black hole mergers to be 10–1700 $Gpc^{-3}$ $yr^{-1}$ and 8–140 $Gpc^{-3}$ $yr^{-1}$, respectively [79].

With the fourth observing run, that will feature the simultaneous engagement of the upgraded interferometers LIGO, Virgo, KAGRA, due to start in late May 2023, big expectation rests in a firm new detection of a nearby binary neutron star merger and its electromagnetic counterpart. For kilonova study purposes, this should clarify whether and how the properties of the single known case, AT2017gfo, will differ from other cases, and how they possibly depend on the gravitational parameters.

A caveat is in order on both the observational and the theoretical front. Firstly, kilonovae are extremely elusive sources, owing to their faintness and fastness. AT2017gfo had a luminosity comparable to that of the least luminous supernovae. It reached peak about one day after the binary neutron star merger, as opposed to supernovae, whose maximum light occurs 10 to 20 days after explosion, and it decayed much faster than a supernova [80]. Therefore, robust kilonova detection requires rapid reactions (seconds to minutes) to transients that should ideally be located at small distances ($z \lesssim 0.1$).

Fast turnaround will be guaranteed by small (meter class) and flexible automatic and robotic telescopes that are already operational or being lined up around the world. Among these is the near-infrared and optical facility Rapid Eye Mount (60 cm) that will continue the GRB-monitoring work it has started since its deployment at the ESO site of La Silla [81,82]. Dedicated optical facilities covering large portions of the sky for optimal GW optical counterpart search include the Gravitational-Wave Optical Transient Observatory, consisting of two arrays of eight telescopes, each with a 40 cm aperture, whose combined field of view is 80 square degrees [83,84]; BlackGEM [85], comprised, in its final configuration, of fifteen 65cm aperture telescopes, with a total field of view of 40 square degrees[1] (see [86] for a complete review of facilities devoted to GW search). Because of low flux levels, spectroscopic identification and characterization requires measurements with 8–10 m class telescopes.

Particularly critical is the rapid search of ultraviolet transients associated with kilonova in its very early phases, when contributions from an ephemeral blue-emission component may be important. These will be efficiently detected by the *Ultraviolet Transient Astronomy Satellite* thanks to its quick-reaction concept and 200-square-degree field of view [87]. During the later phases of kilonova (hours to days after merger) the *James Webb Space Telescope* will provide the most sensitive spectroscopy at infrared wavelengths, potentially also covering the as-yet-unexplored nebular epoch, when the rarefied ejecta allow forbidden emission lines to form. While this phase is observed in supernovae months to years after explosion [88–91], in kilonova it should occur significantly earlier, i.e., weeks after merger, owing to the much smaller ejecta masses.

The future space mission *Transient High Energy Sky and Early Universe Surveyor* for time domain astrophysics, optimized for the search of weak and high-redshift GRBs and currently undergoing a preparatory phase, will presumably have a direct impact on kilonova studies, for it will carry, alongside wide-field X- and gamma-ray cameras, a 70 cm near-infrared telescope to catch prompt kilonova signals associated with short GRBs. Coarse resolution near-infrared spectroscopy will return initial reliable redshift estimates [92].

The source distance, an approximate notion of which can be derived from the gravitational signal, could guide the optical transient searching technique. For nearby sources ($z < 0.1$), it may be more effective to target galaxies selected in the uncertainty area and within the predicted distance range with an algorithm that, based on their size and type, weighs their probability to host a binary compact star system [10,37], rather than using a uniform tiling of the error box, that may be more appropriate when distance is poorly known or not known [93].

From an interpretative point of view, it will be necessary to continue the systematic calculations of line strength for bound–bound transitions and radiative transfer simulations in neutron star merger ejecta for identification of the atomic species. The line strengths depend on the abundance distribution and temperature in the ejecta, which makes ele-

mental identification the most direct diagnostic of the physical conditions of the ejecta. Synthetic spectra, along with line lists, realistic atomic models, and experimentally derived atomic opacities of r-process elements, must be compared with observed kilonova spectra to increase the accuracy of atomic models [63,94]. While the observed spectra of AT2017gfo remains a golden template for modelling work owing to their excellent signal-to-noise ratio, spectroscopic observations of nearby kilonovae hopefully detected during future gravitational observing runs will enable a better discernment of the relative importance of neutron-rich and neutron-poor elements in shaping the kilonova spectrum.

**Funding:** This research was partially funded by the Italian Space Agency and Italian National Institute of Astrophysics under agreement ASI-INAF 2017-14-H.0 High Energy Astrophysics and Astroparticle Physics.

**Data Availability Statement:** The Very Large Telescope X-Shooter spectra of kilonova AT2017gfo are archived in the repository at www.wiserep.org. They can also be obtained from the author on request.

**Conflicts of Interest:** The author declares no conflict of interest.

## Note

1 https://astro.ru.nl/blackgem/, accessed on 16 February 2023.

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
