# Peer review of "Kilonova Emission and Heavy Element Nucleosynthesis"

_universe, doi:10.3390/universe9020105_

Round 1

Reviewer 1 Report

The author presents a brief review of the observations of the kilonova

AT2017gfo associated with the gravitational wave event of binary neutron

star merger GW170817. In view of the opening of the next observing run of

LIGO, Virgo, and KAGRA soon, this short article is a timely account of what

we have learnt from the electromagnetic observations of the single

confirmed kilonova event so far and can serve as a quick summary

for its basic properties. 

The article would be a good supplement to other lengthier reviews which

focus more on the nuclear physics of r-process in compact binary mergers 

(e.g., [48]). I only have a few minor comments for the author to consider. 

1. In Section 3 (Line 66-68), the author may consider adding the following 

more recent work on the r-process of neutron star mergers:

Rosswog et al. A&A 615, A132 (2018)

https://www.aanda.org/articles/aa/full_html/2018/07/aa32117-17/aa32117-17.html

Mendoza-Temis et al. Phys. Rev. C 92, 055805 (2015)

https://journals.aps.org/prc/abstract/10.1103/PhysRevC.92.055805

and also the following recent review for the synthesis of r-process elements (Line 

71-74):

Cowan et al. Rev. Mod. Phys. 93, 015002 (2021)

https://journals.aps.org/rmp/abstract/10.1103/RevModPhys.93.015002

2. Line 78: I’d suggest to use ${}^{56}$Ni instead of ${}^{56}$Nickel.

Author Response

I would like to thank the referee for his/her careful reading of the manuscript and comments.  I have tried to address them,  and specifically:

Referee's point 1:

"In Section 3 (Line 66-68), the author may consider adding the following 
more recent work on the r-process of neutron star mergers:

Rosswog et al. A&A 615, A132 (2018)
https://www.aanda.org/articles/aa/full_html/2018/07/aa32117-17/aa32117-17.html

Mendoza-Temis et al. Phys. Rev. C 92, 055805 (2015)
https://journals.aps.org/prc/abstract/10.1103/PhysRevC.92.055805

and also the following recent review for the synthesis of r-process elements (Line 
71-74):

Cowan et al. Rev. Mod. Phys. 93, 015002 (2021)
https://journals.aps.org/rmp/abstract/10.1103/RevModPhys.93.015002"

Answer: these references were added in the text and in the final bibliography list.

Referee's point 2:

"Line 78: I’d suggest to use ${}^{56}$Ni instead of ${}^{56}$Nickel."

Answer: this was amended.

Again, many thanks to the referee for these suggestions.
Elena Pian

Reviewer 2 Report

The author presents a short review of kilonovae and related phenomena (e.g., gravitational waves and short gamma-ray bursts). The manuscript is well-written and clear, and it will be useful for researchers working in the field.  I have only the following (very minor) remarks: 

- page 2, line 46: "Similarly large sky areas may contain hundreds or thousands of galaxies." 

This statement depends of course on how deep is the image. The number seems too small by several orders of magnitude anyway.

- page 2, line 51: The duration of short GRB is taken as less than 2 seconds typically. Also, short GRBs with extended emission can last much more.

- page 3: line 107: Part of it could also be a simple geometric effect, i.e. due to the gamma radiation emitted from the jet axis, thus delayed with respect to the gravitational wave.

- page 5: line 132: "remain unresolved at radio wavelength". This sentence seems to be in contradiction with the abstract, in which it is said that "For the first time in GRB history, the GRB jet could be angularly resolved in radio VLBI images".  

Author Response

I would like to thank the referee for his/her careful reading of the manuscript and comments.  I have tried to address them,  and specifically:

Referee's point 1:

"
- page 2, line 46: "Similarly large sky areas may contain hundreds or thousands of galaxies." 
This statement depends of course on how deep is the image. The number seems too small by several orders of magnitude anyway.
"

Answer: this sentence was modified and a reference added, see lines 47-50 (updated text is in boldface).

Referee's point 2:

"
- page 2, line 51: The duration of short GRB is taken as less than 2 seconds typically. Also, short GRBs with extended emission can last much more.

Answer: the text was amended; a reference was added, see lines 56-57 (updated text is in boldface).

Referee's point 3:

"
- page 3: line 107: Part of it could also be a simple geometric effect, i.e. due to the gamma radiation emitted from the jet axis, thus delayed with respect to the gravitational wave.
"

Answer: this suggestion was followed and accordingly text added, and a reference added, see lines 118-120 (updated text is in boldface).

Referee's point 4:

"
- page 5: line 132: "remain unresolved at radio wavelength". This sentence seems to be in contradiction with the abstract, in which it is said that "For the first time in GRB history, the GRB jet could be angularly resolved in radio VLBI images".  
"

Answer: I realized and admit that this was confusing.  The abstract, which was imprecise, was accordingly amended, see lines 7-9 (updated text is in boldface).

Again, many thanks to the referee for his/her criticism and suggestions.
Elena Pian

Reviewer 3 Report

This is a nicely written, if rather brief, summary of the observations of GW170817 as they help in understanding the kilonova phenomenon. 

There is some unnecessary repetition about spectral confusion line blending etc in the text around lines 92 and 145, which can probably be shortened. If there is space for more text in the planned special volume, I would find it interesting to learn a bit more about the spectral observations, particularly the way that one can distinguish between neutron-rich and neutron-poor phases and how that relates to the remnant neutron star. But I accept that this is a very technical issue.

One typo is at the start of line 112, where the word is "siren", not "serene". And I think it would be appropriate to refer to the discovery paper of LIGO-Virgo at this point (reference 1), where their distance measurement is being referred to.

Author Response

I would like to thank the referee for his/her careful reading of the manuscript and comments.  I have tried to address them,  and specifically:

Referee's point 1:

"
There is some unnecessary repetition about spectral confusion line blending etc in the text around lines 92 and 145, which can probably be shortened. If there is space for more text in the planned special volume, I would find it interesting to learn a bit more about the spectral observations, particularly the way that one can distinguish between neutron-rich and neutron-poor phases and how that relates to the remnant neutron star. But I accept that this is a very technical issue.
"

Answer: I have removed the repetitions, streamlined the text, and expanded the description of expected kilonova spectra continuum in view of two possible contributions from the neutron-rich (lanthanides) dynamical ejecta and the neutron-poor postmerger ejecta; some references
were also added.  The resulting changes (Sections 3 and 6) are at the lines 92-102 and at the lines 167-173 (updated text is in boldface).

Referee's point 2:

"
One typo is at the start of line 112, where the word is "siren", not "serene". And I think it would be appropriate to refer to the discovery paper of LIGO-Virgo at this point (reference 1), where their distance measurement is being referred to.
"

Answer: this was amended.

Again, many thanks to the referee for his/her criticism and suggestions.
Elena Pian